# In Vitro Efficacy of Isobutyl Cyanoacrylate Nanoparticles against Fish Bacterial Pathogens and Selection Preference by Rainbow Trout (*Oncorhynchus mykiss*)

**DOI:** 10.3390/microorganisms11122877

**Published:** 2023-11-28

**Authors:** Mawuko G. Ahiable, Kouki Matsunaga, Mao Hokin, Kazuhiro Iida, Fumiaki Befu, Syun-Ichirou Oshima

**Affiliations:** 1Laboratory of Cell Structure and Function, Division of Marine Bioresource Science, Graduate School of Kuroshio Science, Kochi University, Nankoku Kochi 783-8502, Japan; ga.mawuko@gmail.com (M.G.A.); k_matsunaga@kyoritsuseiyaku.com (K.M.); m-hokin@kochi-u.ac.jp (M.H.); 2Chikami Miltec Inc., 1-6-3 Ohtesuji, Kochi City 780-0842, Japan; kiida@c-miltec.co.jp (K.I.); fbefu@c-miltec.co.jp (F.B.)

**Keywords:** antibacterial efficacy, isobutyl cyanoacrylate nanoparticles, rainbow trout, safety

## Abstract

The upsurge in havoc being wreaked by antibiotic-resistant bacteria has led to an urgent need for efficacious alternatives to antibiotics. This study assessed the antibacterial efficacy of two isobutyl cyanoacrylate nanoparticles (iBCA-NPs), D6O and NP30, against major bacterial pathogens of fish. In vivo tests on rainbow trout were preceded by in vitro tests of minimum inhibitory concentration (MIC) and minimum bactericidal concentration (MBC). NP30 exhibited higher efficacy than D60, but both iBCA-NPs demonstrated dose-dependent and species-specific in vitro antibacterial properties against the bacterial isolates. Generally, Gram-negative bacteria were more resistant to the iBCA-NPs. *Streptococcus iniae*, *Tenacibaculum maritimum*, and *Photobacterium damselae* were particularly sensitive to both iBCA-NPs. Administered to rainbow trout at 3571.4 mg (iBCA-NP)/kg feed, the iBCA-NPs produced a relative gain rate and survival rates comparable to the control (*p* > 0.05). The condition factor and the hepatosomatic and viscerosomatic indices of fish were indifferentiable (*p* > 0.05) between the iBCA-NP groups and the control. The iBCA-NPs caused no alteration in stress, oxidative stress (superoxide dismutase, SOD), plasma complement titer, or lysozyme activity. This study presents the first report of antibacterial activity of iBCA-NPs against Gram-negative bacteria. The results of this study suggest that D60 and NP30 may contribute to reducing the amounts of antibiotics and chemotherapeutic agents used in aquaculture.

## 1. Introduction

Aquaculture has become very important in increasing global access to fish [1,2] and improving nutrition for the deprived populations of the world [3]. However, the burden of infectious diseases has been identified as the greatest of all setbacks to sustainable global aquaculture growth [4,5]. Losses due to infectious diseases in aquaculture exceed USD 6 billion annually [6]. More than 50% of the most common devastating infectious diseases of fish are caused by bacteria [7,8]. 

In response, frequent and indiscriminate use of antibiotics has characterized aquaculture in many countries [9,10,11]. The consequences have been antibiotic resistance in several bacterial strains and many diverse bacteria in aquacultural systems [12]. As a result, treatments for some common fish bacterial diseases have become less successful, as reported for furunculosis and Edwardsiellosis, increasing losses due to these diseases. Also, some fish bacterial pathogens, including *Streptococcus iniae* and *Photobacterium damselae*, which are zoonotic, carry antimicrobial-resistant genes like extended-spectrum beta-lactamases. These bacteria and/or their resistance genes are transferrable to humans through contact and food [13]. Due to this, some countries have implemented regulations reducing the application of chemotherapeutic drugs in aquaculture [14,15]. Vaccines and probiotics, amongst others, are considered safer alternatives for preventing diseases in aquaculture, but there remain difficulties [16,17]. The use of nanoparticles of high antimicrobial ability is considered the most modern, advanced, and promising approach to fish disease control in aquaculture [18]. Several metal nanoparticles have strong antibacterial properties [19]. However, bioaccumulation in fish flesh, gene toxicity, and the induction of oxidative stress in fish have raised serious concerns about their use in aquaculture [20,21,22]. Preference is growing for nanoparticles that are biodegradable, easy to prepare, and have no or very minimal toxicity threats [23]. Poly (alkyl cyanoacrylate) (PACA) has been used in medicine as an adhesive and its nanoparticles as drug carriers for several decades [24,25,26]. However, the antibacterial properties, and hence the suitability of cyanoacrylate nanoparticles as antibiotic agents, are recent discoveries [27]. Cyanoacrylate nanoparticles have a high affinity for the glycopeptides that make up the cell walls of bacteria. They attach to the cell walls and induce the distortion of cell wall synthesis, leading to autolysis and lysis through mechanisms detailed by Shirotake [23]. Widyaningrum et al. [28] also reported a similar interaction between isobutyl cyanoacrylate nanoparticles, iBCA-NPs, and algae. Thus, unlike antibiotics, iBCA-NPs, like other nanoparticles, employ physical action against pathogens, reducing the possibility of the emergence of resistant bacteria. Their advantages, including easy fabrication and functionalization, biocompatibility, and biodegradability, make them promising candidates for disease control, including fish disease management in the aquaculture industry [26,29]. The antibacterial activity of cyanoacrylate nanoparticles remains a new research area. As far as we know, the application of cyanoacrylate nanoparticles in disease control in fisheries has not been considered by anyone else yet. In this study, we investigated the in vitro antimicrobial activity of two iBCA-NPs (NP30 and D60) against twelve prominent fish disease pathogens. We also assessed the selection preference and safety of D60 and NP30 for the land-based aquacultural production of rainbow trout, *Oncorhynchus mykiss* (Walbaum, 1792), an important aquaculture species, when administered through the feed. 

## 2. Materials and Methods

### 2.1. Isobutyl Cyanoacrylate Nanoparticle Preparation

Two isobutyl cyanoacrylate nanoparticles, iBCA-NPs (D60 and NP30) of different sizes, prepared from isobutyl cyanoacrylate monomers, were used in this work. The isobutyl cyanoacrylate monomers were synthesized from Aron alpha 501F (Toagosei, Tokyo, Japan). The iBCA-NPs were prepared in the laboratories of Chikami Miltec Inc. (Kochi, Japan) and are part of a collection of patented products of the inventors (JP-A-2016-56481; JP-A-2017-81852). Their preparation is detailed in Figure 1, following the methods described by Widyaningrum et al. [28] with slight modifications. Briefly, monomers were added dropwise into 0.2% (*v*/*v*) 5 mol/L HCl until a 1.10% (*v*/*v*) concentration was reached under continuous stirring at 600 rpm and 25 °C. Their respective dispersants (polymerization stabilizers) were then added. In the case of D60, dextran 60,000 (041-30525; FUJIFILM Wako Pure Chemical Corporation, Osaka, Japan) was the dispersant, with a final concentration of 1.10% (*w*/*v*). On the other hand, the nonionic surfactant RHEODOL TW-L (Kao Corporation, Tokyo, Japan) and the anionic surfactant NEOPELEX G-15 (Kao Corporation, Tokyo, Japan) were the dispersants for NP30, at final concentrations 1.25% (*v*/*v*) and 1.0% (*v*/*v*), respectively. After stirring for 2 h to polymerize and prevent particle agglomeration, 0.2 M NaOH was added in drops until the pH reached 7.0. The mixtures were stirred for an additional hour to generate solid nanoparticles. The solution was filtered to remove any debris, if present, using a 5 μm pore size Minisart® NML with CA (Sartorius, Göttingen, Germany). The synthesized iBCA-NPs were stored at 2–5 °C until further use.

#### Characterization of the Synthesized iBCA-NPs

The final concentration of the iBCA-NPs was 10 mg/mL. D60 was diluted by a factor of ten (10) before characterization. Characterization analyses were conducted in triplicate on iBCA-NP suspensions. Dispersed diameter values for each iBCA-NP were measured immediately after production using the dynamic light scattering (DLS) method. The effective surface charge (zeta potential) was determined via the electrophoretic light scattering method. At the same time, particle size, particle distribution, and polydispersity index of the particles were also measured via the DLS method using a Zetasizer Nano ZS90 (Malvern Panalytical, Malvern, UK).

### 2.2. Bacterial Pathogens

Aliquots of twelve important fish pathogenic bacterial isolates from eight different genera (*Aeromonas*, *Edwardsiella*, *Nocardia*, *Lactococcus*, *Photobacterium*, *Streptococcus*, *Tenacibaculum*, and *Vibrio*) were tested for their sensitivity to NP30 and D60. These bacterial pathogens were isolated from infected fish in different regions of Japan. All except *Tenacibaculum maritimum* were cultured in Brain-Heart Infusion nutrient broth (BHI, Difco Becton, Dickinson, and Company, Sparks, MD, USA). *T. maritimum* was grown in marine broth (Difco, Becton, Dickinson, and Company, Sparks, MD, USA). They were then cryopreserved at −80 °C in equal volumes of their respective broths and of 40% glycerol at the Laboratory of Cell Structure and Function, Division of Marine Bioresource Science, Graduate School of Kuroshio Science, Kochi University, Nankoku, Kochi, Japan. Identification of the bacteria species included macroscopic examination of the pathological signs of infected fish, characteristic morphology on nutrient agar, Gram-staining, and subsequent molecular confirmation.

Molecular identification of all the bacterial isolates was executed using 63f (5′-CAGGCCTAACACATGCAAGTC-3′) and 1387r (5′-CGGCGGWGTGTACAAGGC-3′) bacterial 16S rRNA primers [30] following Ahmed et al. [31] with minor modifications. Briefly, amplified PCR products were purified using EXOSAP-IT (Thermo Fisher Scientific Baltics UAB, Vilnius, Lithuania) and then subjected to sequencing. Each nucleotide sequence was queried against the NCBI GenBank database by BLAST^®^ [32] in MEGA version X [33], and homologous sequences with ≥97% similarity were selected to identify each bacteria isolate.

#### Bacterial Culture Medium for MIC Assay 

The bacteria isolates included the typical and atypical *Edwardsiella tarda* strains [34,35], *Nocardia seriolae*, *Lactococcus garvieae*, *Photobacterium damselae* subsp. *piscicida*, *Streptococcus iniae*, *T. maritimum,* and four *Vibrio* spp. (*V. anguillarum*, *V. harveyi*, *V. parahaemolyticus*, and *V. rotiferanus*). All bacteria strains except *T. maritimum* were cultured in Tryptic Soy Broth, TSB (Bacto, Becton, Dickinson, and Company, Sparks, MD, USA) in 2% NaCl for all except for the two *E. tarda* strains and *A. salmonicida* subsp. *salmonicida* (where NaCl = 0.5%). They were incubated in a shaking incubator at 100 rpm except for *A. salmonicida* subsp. *salmonicida* (120 rpm) and *N. seriolae* (180 rpm). The incubation temperature was 25 °C, except for the two *E. tarda* strains (28 °C). *T. maritimum* was cultured in marine broth (Difco, Becton, Dickinson, and Company, Sparks, MD, USA) and incubated at 25 °C, 180 rpm. The growth conditions for the bacterial strain followed the Revised Standard Methods of the Japanese Society of Antimicrobials for Animals (JSAA) for antimicrobial tests against bacteria isolated from Animals [36].

### 2.3. MIC Measurement by Micro Liquid Dilution Method

Susceptibility tests were carried out by the standard broth microdilution method according to the performance standards for antimicrobial susceptibility testing by the Clinical Laboratory Standards Institute [37]. The minimum inhibitory concentration (MIC) of the iBCA-NPs was evaluated spectrophotometrically by the turbidimetric method. The overnight (or longer, 48 h for *T. maritimum* and 96 h for *N. seriolae*) bacterial cultures were double-fold serially adjusted to 1.0 × 10^5^ CFU/mL in 1.5 mL Eppendorf tubes (Wako, Osaka, Japan). A total of 100 μL of 10 mg/mL each of NP30 and D60 was double-fold serially diluted up to at least 0.078 mg/mL in 96-well plates and inoculated with the 1.0 × 10^5^ CFU/mL bacteria inoculums. Blank controls consisted of 100 μL inoculum-free fresh broth + iBCA-NPs mixture, while inoculated 100 μL nanoparticle-free fresh broth served as the positive control. The method was slightly modified for *N. seriolae*. The cells of *N. seriolae* form large aggregates in broth and firmly stick to surfaces like the 1.5 Eppendorf tubes and pipette tubes, making it challenging to inoculate the assay using the CLSI [37] prescribed method successfully. Therefore, for this bacterium, the MIC assay used was 100 μL of bacterial inoculums adjusted to 1.0 × 10^5^ CFU/mL in fresh broth inoculated into 100 μL of iBCA-NPs serially diluted with sterilized distilled water (total assay volume = 200 μL). The plates were incubated in a static incubator at recommended standard conditions of temperatures and time, as indicated above. Optical densities at 620 nm (OD_620_) absorbance were measured for each MIC assay before and after incubation. If the MIC assay was more turbid after incubation, bacterial growth was implied, while lower or similar OD_620_ values to before incubation indicated growth inhibition [38]. The lowest concentration at which each iBCA-NP inhibited at least 80% growth of the test bacteria was considered the related MIC value [39], deduced from plots, as exemplified in Figure 2. The procedure was also executed with the dispersants of each iBCA-NP, as shown in Figure 2b, to assess the source of antibacterial activity and the role of the dispersants. Each assay was prepared under aseptic conditions and performed in triplicate, and the result that occurred twice or more was considered [31,40]. Percentage bacterial growth inhibition (GI %) was computed using the following formula:(1)$$\mathrm{Growth}\;\mathrm{Inhibition}\;\mathrm{rate}\;(\mathrm{GI}\%)=1-\left(\frac{\mathrm{OD}\;\mathrm{of}\;\mathrm{sample}}{\mathrm{OD}\;\mathrm{of}\;\mathrm{control}}\right)\times 100$$

The GI % (showing the dose-dependent and species-specific antibacterial efficacy of each iBCA-NPs) at each concentration of the iBCA-NPs was determined and expressed as mean ± standard error of the mean (SEM) in bar graphs (Figure 3, Figure 4 and Figure 5).

#### Minimum Bactericidal Concentration (MBC) Assay

The MBC corresponding to each sensitive bacteria was the lowest concentration of each iBCA-NP, which inhibited 99.9% of the bacterial growth on suitable growth agar plates of the bacteria strain [41,42]. To determine the MBC, 50 μL of the bacteria-iBCA-NP mixture at the MIC and higher concentrations were plated on suitable nutrient agar for each bacteria strain. The plates were then incubated in a static incubator at recommended conditions, as stated above. The absence of bacterial colonies on plates after the recommended incubation time [36] signified the lack of living bacteria and the bactericidal effect at that concentration. On the other hand, the growth of bacterial colonies on plates indicated the presence of living bacteria and, hence, a bacteriostatic effect at that concentration. The assay was performed in triplicate, and results that were repeated twice or more were considered.

### 2.4. Selection Preference and Growth Performance of iBCA-NPs-fed Rainbow Trout

In vivo tests are the ultimate determinants of the relevance of the iBCA-NPs, even if found to be efficacious in vitro. We found that immersion in 100 ppm and higher concentrations of the iBCA-NPs resulted in fatalities of active swimmers like the rainbow trout, red seabream (*Pagrus major*, (Temminck & Schlegel, 1843)), and the Japanese amberjack (*Seriola quinqueradiata*, (Temminck & Schlegel, 1845)), although passive swimmers like the olive flounder (*Paralichthys olivaceus*, (Temminck & Schlegel, 1846)) survived (unpublished data). Dose-dependent reduced feed intake and mortality characterize the oral administration of some antibiotics to fish [42]. Long-chain PACA and its nanoparticles are considered safe for body tissues and can be efficiently eliminated from the body by excretion [25,26,27]. We therefore tested the selection preference, growth performance, and safety of NP30 and D60 when administered orally through feed to rainbow trout. The basal diet was EP commercial feed grade 4 (Nisshin Marubeni Feed Co., Tokyo, Japan) for rainbow trout. Rainbow trout fry raised at a seed and nursery center in Ehime Prefecture were transported to the Freshwater Experiment Site of the Kochi University (Kochi, Japan), where they were given the EP commercial feed in a 1.1-ton fiberglass reinforced plastic (FRP) tank until the experiments began. 

#### 2.4.1. Experimental Design

The experimental setup included six flow-through 500 L FRP tanks demarcated into three groups, two replicates each (*n* = 2). Each tank contained 45 fish (mean weight 35.08 ± 0.46 g, mean length 14.18 ± 0.09 cm). The iBCA-NP groups (NP30 and D60 groups) were fed EP feed supplemented with the respective iBCA-NP solutions at the maximum amount that could be added to the feed, which was determined to be 3571.4 mg(iBCA-NP)/kg feed. The control groups were fed EP feed mixed with the same volume of distilled water (as the iBCA-NPs). The fish were acclimatized in their respective new tanks on iBCA-NP-free basal feed for two weeks before the start of treatments. The water was degassed and well aerated, and the volume was kept at 400 L for each tank. Tanks were maintained under natural light/day regimes. Water quality parameters were frequently recorded throughout the study, where the water exchange rate was 85.5% per hour per tank, the mean temperature was 18.35 ± 0.05 °C, pH was 8.38 ± 0.17, and DO was 9.80 ± 0.21 mgL^−1^. Fish in each tank were fed to satiation twice daily (morning and evening) for the study period (12 weeks). Daily feed intake per tank and feeding behavior were recorded.

#### 2.4.2. Sampling

Five (5) fish samples were randomly taken from each tank every 28 days and anesthetized with a lethal dose of 2-phenoxyethanol (Wako). Then, 1 mL blood from the caudal vein of each fish sample was aliquoted into labeled 1.5 mL Eppendorf tubes using a heparinized syringe and kept on ice. The blood samples were centrifuged at 3000× *g* for 15 min at 4 °C to obtain plasma. The blood plasma collected was frozen at −80 °C until use. Morphometrics (weight and length) were also measured for all groups every 28 days for the study period (12 weeks).

#### 2.4.3. Growth Performance

The following equations were used to calculate the growth performance parameters:(2)$$\mathrm{Weight}\;\mathrm{gain}\;(\mathrm{WG}\%)=\frac{\mathrm{final}\;\mathrm{weight}\;\left(g\right)-\;\mathrm{initial}\;\mathrm{weight}\;\left(g\right)}{\mathrm{initial}\;\mathrm{weight}\;\left(g\right)}\times 100$$
(3)$$\mathrm{Specific}\;\mathrm{growth}\;\mathrm{rate}\;(\%\;\mathrm{per}\;\mathrm{day})=\frac{\mathrm{Ln}\;\left(\mathrm{final}\;\mathrm{weight}\right)-\;\mathrm{Ln}\;\left(\mathrm{initial}\;\mathrm{weight}\right)}{\mathrm{duration}\;\left(12\;\mathrm{days}\right)}\times 100$$

#### 2.4.4. Blood Plasma Lysozyme Activity and Complement Titer 

Lysozyme activity in blood plasma was examined via the turbidimetric assay and lysis of the lysozyme-sensitive Gram-positive bacterium *Micrococcus lysodeikticus* (Sigma), according to [43]. For complement titer, sample plasma and 25 µL of sheep red blood cells (Japan Lamb) were suspended and allowed to react for 1 h at room temperature. After completion of the reaction, 50 µL of EGTA, Mg^2+^, and GGVB (glucose gelatin veronal buffer with 10 mM glycol ether diamine tetra acetic acid and 40 mM MgCl_2_) were added, and the mixture was centrifuged at 4 °C for 3000× *g* for 3 min. Then, 70 µL of the supernatant was added to a 96-well plate, and the absorbance value at 493 nm wavelength was measured with a microplate reader (Thermo Scientific Multiskan FC, Shanghai, China). The complement titer, y, was computed using the following formula: Complement titer y = {absorbance value of sample − CB (mechanical hemolysis, 25 µL of red blood cells + 75 µL of buffer solution)/(absorbance value of 100% hemolysis − absorbance value of CB)} × 100(4)

#### 2.4.5. Stress and Oxidative Stress Biomarkers

Stress affects the sensitivity of the immune response to stimuli and the occurrence of infectious disease in fish [44]. Pollutants significantly affect fish condition factor, the state of fish liver, and the viscera of fish [45,46,47]. We estimated the condition factor, hepatosomatic index, and viscerosomatic index of fish in each treatment to ascertain fish health and condition. Indications of possible acute stress consequent to the iBCA-NP treatments were investigated via glucose concentration, which was found to rise with rising cortisol levels in response to stress in fish [48,49]. The glucose concentration in blood sera was measured at OD_493_ using the Mutarotase-GOD method according to Miwa et al. [50] via a commercially available Glucose C-II kit following (FUJIFILM Wako Pure Chemical, Osaka, Japan). Superoxide dismutase (SOD) activity in blood plasma was spectrophotometrically estimated at 493 nm by the xanthine oxidase method using the SOD Assay Kit-WST (Dojindo, Kumamoto, Japan) following the manufacturer’s guideline.

Our work strictly followed all the ethical protocols for experimenting with aquatic organisms as stipulated by Kochi University’s Animal Experiment Committee, which gave ethical clearance for this work (Ethical Clearance Code QA-02).

### 2.5. Statistical Analysis

Results of the antimicrobial tests were produced in triplicate and are presented as mean ± standard error of the mean (SEM, *n* = 3). Data from the in vivo growth test were subjected to statistical verification using one-way ANOVA tests at a 95% confidence level in IBM SPSS for WINDOWS (version 29.01.0). Tukey’s post hoc test was used to confirm differences where differences occurred within the experimental groups. Data that violated the parametric assumptions were analyzed using the Kruskal–Wallis tests. We also computed the achieved statistical power of our samples using a two-tailed *t*-test in G*Power for WINDOWS (version 3.1.9.7), according to Faul et al. [51,52].

## 3. Results 

### 3.1. Physicochemical Characteristics of iBCA-NPs

Analysis of the zeta potential of the two iBCA-NPs showed that both D60 and NP30 were negatively charged (NP30 = −20~−25 mV; D60 = −5~−10 mV). There was no significant difference between concentrations. The mean diameter of the iBCA-NPs ranged from 150–180 nm for D60 and 10–50 nm for NP30. The polydispersity index (PDI) and size distribution analyses showed that particles of both D60 and NP30 were largely monodispersed, with a uniform and relatively narrow size distribution (PDI (D60) = 0.15 to 0.45; PDI (NP30) = 0.05–0.30).

### 3.2. Antibacterial Effect of iBCA-NPs

#### 3.2.1. Minimum Inhibitory Concentration (MIC)

Both isobutyl cyanoacrylate nanoparticles (NP30 and D60), but not their dispersants (Figure 2 and Appendix A), exhibited species-specific and dose-dependent antibacterial activity against the tested fish bacteria pathogens (Table 1; Figure 3, Figure 4, Figure 5 and Figure 6). However, NP30 was more efficacious, showing antibacterial activity against all twelve tested bacteria strains and at lower concentrations (Table 1; Appendix A). At the highest concentration, 5 mg, D60 was also efficacious against all except *V. harveyi* (Figure 3b and Figure 6a). *S. iniae* and *L. garvieae* (Gram-positive), *P. damselae*, and *T. maritimum* (Gram-negative) were the most sensitive species (MIC = 78 or 156 µg/mL for both iBCA-NPs). Three *Vibrio* spp. (*V. harveyi*, *V. parahaemolyticus*, and *V. rotiferanus*) and *A. salmonicida* subsp. *salmonicida* demonstrated the highest resistance. MIC was not obtained for D60 against *V. harveyi* at the iBCA-NP concentrations assessed.

#### 3.2.2. Minimum Bactericidal Concentration (MBC)

MBC referred to the lowest concentration of each iBCA-NP at which no bacteria growth was observed on nutrient agar plates. NP30 was bactericidal against all tested bacterial isolates, while D60 was bactericidal against only two isolates, namely, *T. maritimum* and *V. anguillarum*. D60 was, thus, dominantly bacteriostatic against the test bacteria strains (Table 1). *T. maritimum* recorded the lowest MBC for both iBCA-NPs. 

The extent of dose-dependent percentage growth inhibition (GI %) demonstrated by each iBCA-NP was reported in Figure 3, Figure 4 and Figure 5. Consistent with our MIC and MBC results, generally, NP30 showed higher rates of bacterial growth inhibition against most bacterial isolates.

### 3.3. Growth Performance and Safety Evaluations in Rainbow Trout

The iBCA-NPs had no adverse effect on water quality parameters throughout the studies. The mean temperature was 18.35 ± 0.05 °C, pH was 8.38 ± 0.17, DO was 9.80 ± 0.21 mgL^−1^, total ammonium was 0.42 ± 0.05 mgL^−1^, nitrite was 0.02 ± 0.003 mgL^−1^, and nitrate was 0.35 ± 0.04 mgL^−1^. The survival rate was highest (100%) in the NP30 group but not significantly different (*p* > 0.05) from the others (98.89%). Feeding aggression was relatively higher in the iBCA-NP groups than in the control. After 12 weeks of feeding, rainbow trout fed with iBCA-NPs (both NP30 and D60) supplemented feed showed relative gain rate, specific growth rate, and feed conversion ratio (FCR) as good as (*p* > 0.05) fish in the control group fed the commercial EP feed only. The relative gain rate (weight gain rate) was steeply high for the first month but significantly declined (*p* < 0.001) in subsequent months for all groups. (Figure 7a). Generally, NP30-treated tanks showed slightly better growth performance, but there was no significant difference among the treatments (Table 2; *p* > 0.05). At the end of 8 weeks, FCR was significantly lower (*p* < 0.05; Appendix A) in the iBCA-NPs groups than in the control. However, FCR was not differentiable (*p* > 0.05) among treatments at the end of the study.

### 3.4. Innate Defense Capability 

Macroscopic physical examinations of fish during each sampling time suggested healthy growth in all treatments. Lysozyme activity increased with the age of rainbow trout. The control group had the highest lysozyme activity (0.092 ± 0.031 U/mL) at the end of the eighth week but declined steadily to be the lowest (0.072 ± 0.01 U/mL) at the end of this study. On the other hand, lysozyme activity of the D60 group was the lowest (0.017 ± 001 U/mL) at the end of the fourth week but steadily increased to become the highest (0.089 ± 0.025 U/mL) at the end of the 12th week (Figure 7b). Lysozyme activity after the first four weeks was significantly lower (*p* < 0.05) than the activity rate after the twelfth week (Figure 7b). However, there was no significant difference among the treatments at any sampling time (*p* > 0.05).

The control groups had the lowest complement titer after the first month (mean 0.158 ± 0.017 U/mL), while NP30 always had the highest hemolysis rate, reaching 0.408 ± 0.023 U/mL at the end of eight weeks. Like lysozyme activity, complement titer values were significantly lower (*p* < 0.05) for all groups (mean = 0.174 ± 0.02 U/mL) at the end of the fourth week than after eight weeks (mean = 0.342 ± 0.033 U/mL). However, there was no significant difference in hemolysis rate among treatments at any sampling time (*p* > 0.05). 

### 3.5. Stress-related Biomarkers

Condition factor, hepatosomatic, and viscerosomatic indices were consistently similar among groups and not significantly different (*p* > 0.05; Table 2). At the end of the first four weeks, glucose concentration was slightly higher in the D60 group, but glucose concentrations were strikingly similar for all treatments at the end of the study (Figure 7d). The overall rate of SOD activity was slightly higher in the iBCA-NP groups (control = 67.64 ± 0.37; D60 = 68.60 ± 0.53; NP30 = 69.84 ± 0.43%), but there was no significant difference among treatments (Figure 7e). However, we found one fish with a severe internal tumor in an NP30 tank (Figure 8). The statistical power for our monthly samples per treatment was stronger in months one and two than in month three. Generally, the power varied widely from very high (99.99%) for first-month lysozyme concentration to very low (5.58%) for the third month estimates of plasma complement titer. The mean achieved statistical power for our monthly samples per treatment was 50.62 ± 8.59%. 

## 4. Discussion

The characteristics of the cyanoacrylate nanoparticles, iBCA-NPs (D60 and NP30), were very similar to iBCA-NPs (25 nm and 180 nm, respectively) described by Widyanigrum et al. [28]. The similarities were because the monomers used were the same, and the reagents and formulation methods were very similar.

From the results of the antibacterial activity and bactericidal activity examined in this study, the iBCA-NPs we studied demonstrated intense antibacterial activity against all the bacterial pathogens we examined (Table 1; Figure 3, Figure 4, Figure 5 and Figure 6). NP30 was bactericidal against 100% of the pathogens, while D60 was bacteriostatic against most (Table 1). Shirotake [27] suggested that the mechanism of antibacterial activity of iBCA-NPs includes high affinity of the iBCA-NPs for glycoproteins, resulting in junctions between iBCA-NPs and Gram-positive bacteria, leading to bacterial wall autolysis. For algal cells, iBCA-NPs with small particle size (e.g., 25 nm) induced reactive oxygen species generation and cell mortality [28]. The dispersant types played crucial roles in differentiating between the sizes of the two iBCA-NPs and prevented their agglomeration, consistent with Guzman et al. [53]. However, the dispersants did not interfere with the antimicrobial effect of our iBCA-NPs, as none showed significant antibacterial activity against the bacterial strains (Figure 2 and Appendix A). That is, from our results, the iBCA-NPs, not the dispersants, were the source of the antibacterial activities observed. Widyaningrum et al. [28] reported a similar finding, where the dispersants did not affect algal cell walls. The massive global aquacultural losses to these bacteria species are well documented [54,55,56,57,58,59,60,61,62]. The cost of using antibiotics to control these pathogens is enormous [11], and it will require several different types of vaccines to combat them. The importance of the efficacy of one safer antimicrobial agent, iBCA-NP, to manage several bacterial strains cannot be overemphasized.

The comparatively higher antimicrobial activity of NP30 might be due to its minute particle size (mean size = 30 nm), increasing the surface area for attachment to the bacterial pathogens. This assertion agrees with Widyaningrum et al. [28], who found an inverse relationship between iBCA-NPs size and algal cell mortality at any given concentration. D60 particles were larger (mean size = 180 nm), resulting in a more viscous colloid (Figure 1b). The superior performance of NP30 might also be explained by the negative electrical charge effect between NP30 particles induced by its anionic dispersant. This effect is thought to keep particles uniformly distributed in solution increasing contact with bacteria cells. The nanoparticle size–antibacterial activity relationship has been widely studied. Consistent with our results, generally, the smaller the size, the higher the activity [63,64,65].

Lipopolysaccharide outer membrane and chromosomally encoded drug efflux mechanisms in Gram-negative bacilli confer comparatively higher antibacterial resistance on them than on Gram-positive bacteria [66,67,68]. Consistent with this, Gram-negative bacteria were more resistant to our iBCA-NPs (Table 1). Notably, although the susceptibility of several Gram-positive bacteria, including multidrug-resistant bacteria, to iBCA-NPs has been reported, all previous work reported no antibacterial activity of iBCA-NPs on Gram-negative bacilli [27,69]. The extreme sensitivity of *P. damselae* and *T. maritimum* and, indeed, the susceptibility of other Gram-negative bacilli we tested, was, thus, an exciting result. Differences in the iBCA-NPs preparation method and bacterial strains used might have accounted for the variations in results. It is known that differences in structure and composition between N-acetylglucosamine and N-acetylmuraminic acid may account for differences in antimicrobial resistance among Gram-negative bacteria [70,71]. The antimicrobial resistance of *Aeromonas* spp. and *Vibrio* spp., consistent with our results, are well documented [31,72,73,74,75,76,77]. On the other hand, although differences have been demonstrated in the mechanism of virulence and pathogenicity of the typical and atypical *E. tarda* strains [34,35], their antimicrobial susceptibility and resistance to both iBCA-NPs were very similar (Table 1). Their structural indistinguishability and molecular closeness might have accounted for the observation [78,79].

We observed increasing GI % with decreasing concentration for NP30 against *T. maritimum* (Figure 5c), which we initially thought to be the display of the Eagle effect [39]. Divya et al. [80] reported similar results, where lower, rather than higher, concentrations of some of their chitosan nanoparticles showed higher growth inhibition. However, bactericidal properties were observed at higher NP30 concentrations (5 mg) and lower concentrations (until 0.313 mg). *T. maritimum* is a filamentous bacterium with rhizoids with which gliding colonies adhere to one another, creating a massive network of bacteria mass [58]. On the other hand, the iBCA-NPs assay transparency increased (decreasing OD_620_ value) by dilution. The effect of bacteria lysis and assay transparency on OD readings was suspected to have accounted for the inconsistency observed. 

*Growth performance and safety evaluation in rainbow trout*: In our in vivo experiments, water quality parameters were not adversely affect [81] by the iBCA-NPs. In vitro, as well as in vivo, medium-molecular-weight cyanoacrylate nanoparticles like iBCA-NPs significantly degrade in 24 h [24,25,29]. The iBCA-NP biodegradability, coupled with the high rate of water exchange (85.5% h^−1^), might have played a role in maintaining stable water quality parameters despite the heavy feed administration. 

The growth test lasted twelve weeks, which we considered long enough to reveal any effects the iBCA-NPs might have on rainbow trout. During this duration, rainbow trout were consistently fed with iBCA-NPs-supplemented feed at 3571.4 mg/kg, the highest concentration that did not interfere with the integrity of the feed. At high concentrations, several metallic nanoparticles are toxic, killing fish [49]. Growth inhibition was also reported in *Cyprinus carpio* by copper nanoparticles after thirty days of exposure [82]. Similarly, dose-dependent reduced feed intake and fish mortality have been associated with oral administration of some antibiotics [83]. On the contrary, both feeding efficiency and growth performance were as excellent in the iBCA-NPs groups as in the control groups (Table 2, Appendix A), and sometimes slightly better, but not significantly higher (*p* > 0.05), than the latter. 

The comparatively significant decline in weight gain for all treatments, including control groups (Figure 7a), after the first four weeks was consistent with documented growth trends of rainbow trout. Feed conversion efficiency is reportedly lower in older rainbow trout [84]. Fish condition factor, K, hepatosomatic, and viscerosomatic indices (HIS and VSI, respectively) are important indicators of fish health, metabolism, and energy reserves in fish [45,46,47]. The K, HSI, and VSI values obtained in this study are similar to those reported for healthy fish in similar studies [45,85]. Therefore, like the growth performance parameters, these indices suggested no toxic effects of the iBCA-NPs (Table 2).

Both the complement system and lysozymes are essential components of the first barrier of fish against disease pathogens and play crucial roles against the growth of bacterial agents. They also possess lytic activity against viruses and both Gram-positive and Gram-negative bacteria [86,87,88,89]. In addition, the complement system of fish is critical in modulating the adaptive immune response. It has chemotaxic, opsonization, and proinflammatory functions, making it one of the most critical factors in the fish’s innate immune system [86,87,90]. Increased alternative complement hemolysis of foreign objects like sheep erythrocytes indicates improved immune response [86]. While the relationship between the level of lysozyme activity and immunity in fish remains unclear; generally, long-term exposure to toxins and immunosuppressants have decreasing effects on it. On the other hand, immunostimulants and dietary supplements like prebiotics are associated with increasing lysozyme levels and fish complement titer, conferring protection against various fish diseases [87,88,89]. 

In this study, lysozyme activity was significantly higher (*p* < 0.05) in the twelfth week than in the fourth week (Figure 7b). Similarly, the complement titer was significantly higher in the eighth week than in the fourth week (Figure 7c). Generally, the immune indicators increased with age, suggesting improved immunity of rainbow trout with age, irrespective of treatment. This result agrees with Sauer et al. [91] that age improves, not declines, fish health. Considering both indicators, therefore, iBCA-NPs had no adverse effects on rainbow trout in this study.

The primary concerns about nanoparticle use in aquaculture include, among others, oxidative damage to fish and impairment of many biomarkers. Metallic-nanoparticle-induced oxidative stress has been reported by several publications [22,49,92,93,94,95]. Similar damages in fish have been reported using carbon-based and green synthesized cadmium nanoparticles [96,97]. These damages have been proven to correlate with increased acute stress, signified by increased concentrations of blood cortisol and glucose in the Nile tilapia and the African catfish [49,97]. Exposed to toxic effects, the glucose concentration of fish increases due to increased rates of glycolysis in the fight against the toxin [97]. On the contrary, in the current study, glucose concentration for all groups was consistently optimal (mean = 60.02 ± 0.76 mg/dl) [48,49,97]. Therefore at 3571.4 mg(iBCA-NP)/kg feed, the iBCA-NPs did not induce stress in rainbow trout. Compared to the liver, blood plasma estimates of oxidative stress are more accurate because the former has a high activity of antioxidant enzymes to adapt to and cope with oxidative stress [94]. Persistent or high oxidative stress may cause notable damage to proteins, decreasing SOD activity [94]. On the contrary, plasma SOD activity also confirmed that the SOD antioxidant defense systems of iBCA-NPs treated rainbow trout were at least as good as those of fish in the control group (Figure 7d). 

Several pathways have been predicted for orally administered poly (alkyl cyanoacrylate) nanoparticles, PACA-NPs. According to Vauthier et al. [26], translocation via the Peyer patches in the ileum is the primary pathway for drug-carrier PACA-NPs. Catalyzed by esterases from serum, lysosomes, and pancreatic juice, they are eliminated from the body by the kidneys in the form of alkyl alcohol and poly-cyanoacrylic acid. Unlike shorter-chain derivatives, butyl derivatives and higher homologs of PACA are considered nontoxic to tissues [25] and safer than other antimicrobial substances [27]. In agreement, from the results of the current study, at 3571.4 mg(iBCA-NP)/kg feed, iBCA-NPs produced excellent growth, and the innate immunity of rainbow trout was not compromised. It was difficult to account for the individual with a severe tumor found in one NP30 tank because it was an isolated case. In addition, it seems unlikely that the tumor was caused by the iBCA-NP because the dominant medical use of cyanoacrylate nanoparticles is in the fight against tumors and cancers [98,99,100,101]. However, the possibility cannot be completely ruled out because little is known about the dose-dependent effects of the surfactants for NP30. The moderate to low overall achieved statistical power of our monthly samples may be considered a limitation of our study. A higher sample size is therefore recommended for future studies [102].

## 5. Conclusions

This study presents the first report of antibacterial activity of iBCA-NPs against Gram-negative bacteria. Both D60 and NP30 demonstrated high in vitro antibacterial and bactericidal and or bacteriostatic properties against both the Gram-positive and Gram-negative bacterial pathogens of fish diseases we tested. In vivo, we confirmed the safety of D60 and NP30 for fish culture by land-based aquaculture using rainbow trout, an industrially important salmonid species. It is expected that these results will be replicated irrespective of the system of production. However, in vitro efficacy of antimicrobials may only sometimes be replicated in vivo. Given the favorable selection of the iBCA-NPs by rainbow trout and indications of their safety to fish, we recommend tests to investigate if the iBCA-NPs are effective as prophylactic agents or therapeutic in case of bacterial infections in fish. 

## Figures and Tables

**Figure 1 microorganisms-11-02877-f001:**
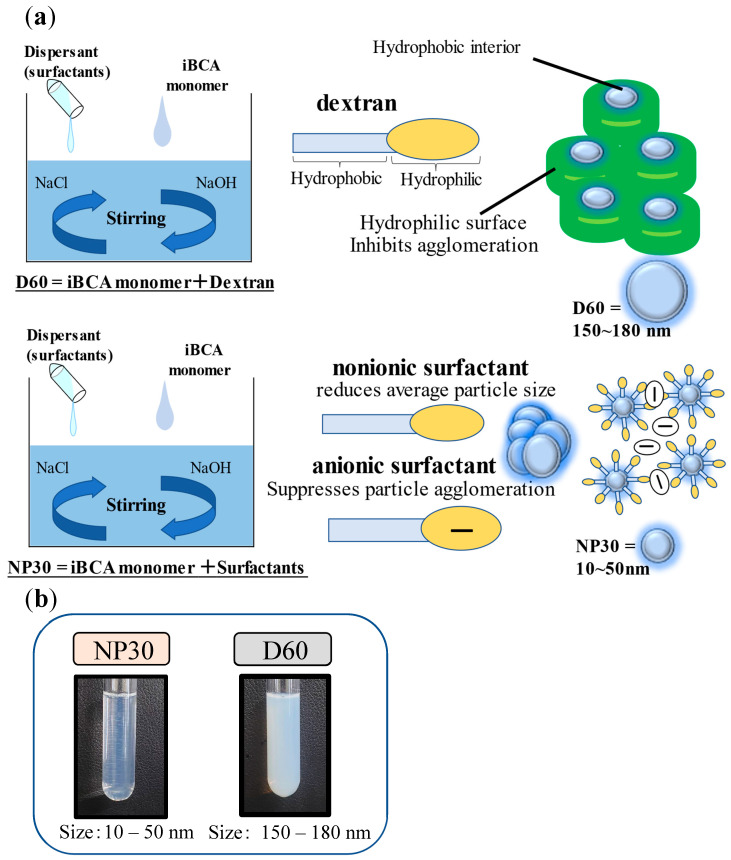
(**a**) Preparation of the isobutyl cyanoacrylate nanoparticles (iBCA-NPs) used in this study. (**b**) Plate comparing cloudiness of the iBCA-NP solutions used in this study.

**Figure 2 microorganisms-11-02877-f002:**
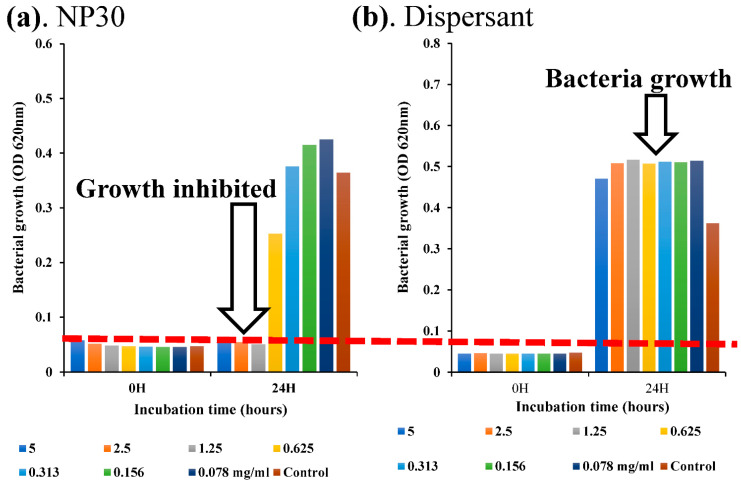
A plot to determine MIC, showing bacterial growth inhibition after 24 h in NP30 (**a**) vs. bacterial growth in dispersant (**b**). From the plot, the MIC of NP30 against *E. tarda* (*typical*) is 1.25 mg/mL.

**Figure 3 microorganisms-11-02877-f003:**
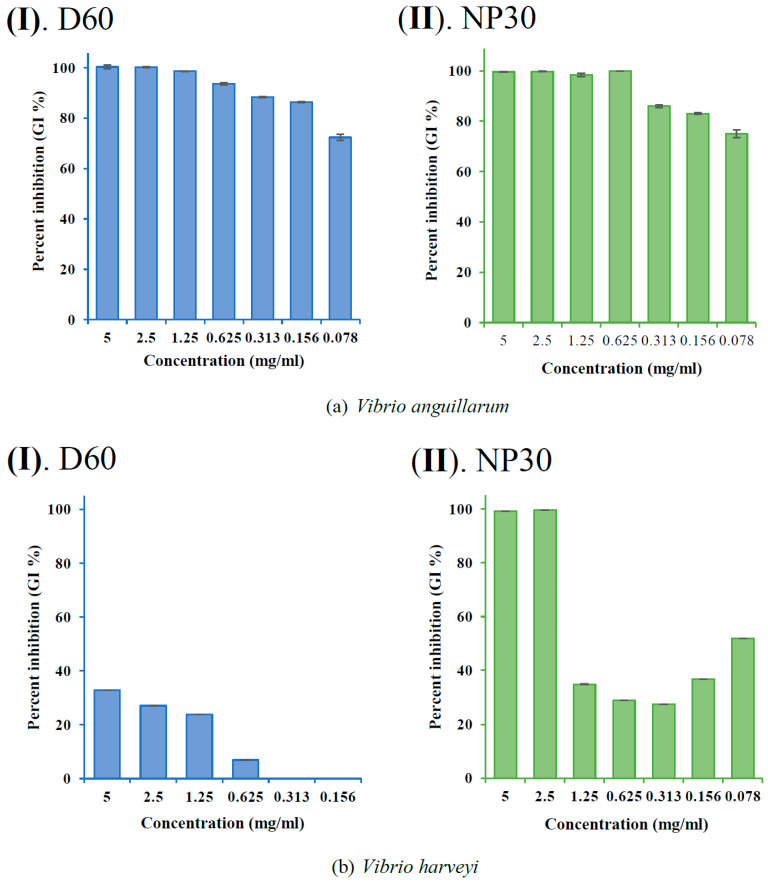
Percent growth inhibition (GI %) of *Vibrio* spp. at all growth inhibiting concentrations: (**a**) GI % of *Vibrio anguillarum* for D60 (I) and NP30 (II); (**b**) GI % of *Vibrio harveyi* for D60 (I) and NP30 (II); (**c**) GI % of *Vibrio parahaemolyticus* for D60 (I) and NP30 (II); (**d**) GI % of *Vibrio rotiferanus* for D60 (I) and NP30 (II). All data are expressed as means ± SEM (*n* = 3).

**Figure 4 microorganisms-11-02877-f004:**
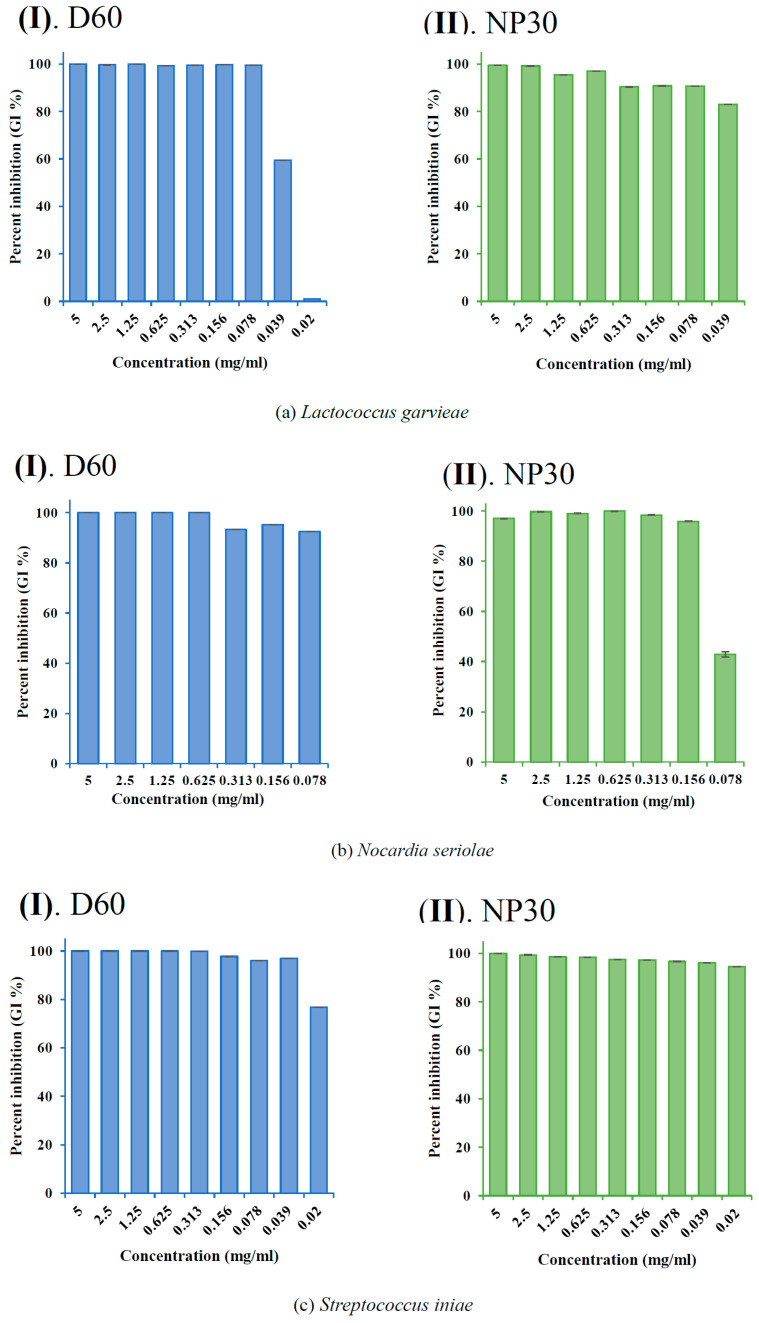
Percent growth inhibition (GI %) of some Gram-positive bacteria strains at all growth-inhibiting concentrations observed: (**a**) GI % of *Lactococcus garvieae*; (**b**) GI % of *Nocardia seriolae*; (**c**) GI % of *Streptococcus iniae* for D60 (I) and NP30 (II). All data are expressed as means ± SEM (*n* = 3).

**Figure 5 microorganisms-11-02877-f005:**
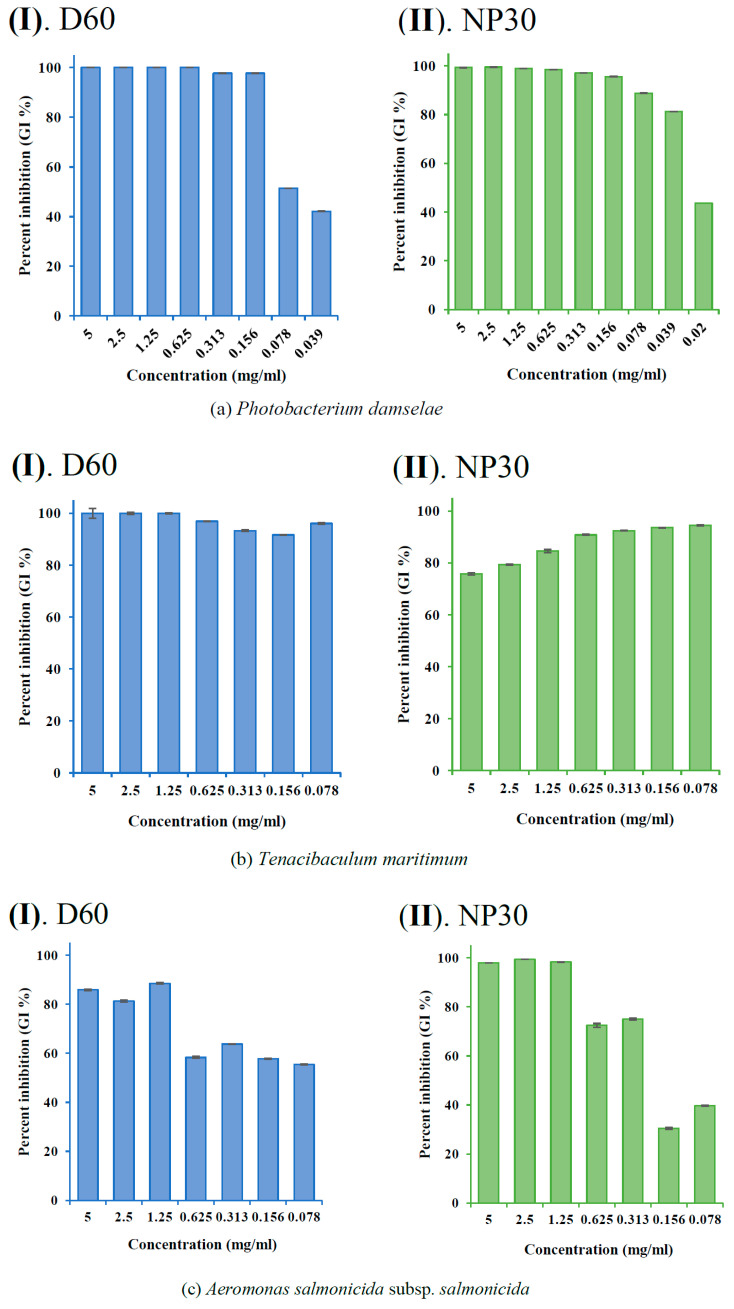
Percent growth inhibition (GI %) of some Gram-negative bacteria strains at all growth-inhibiting concentrations observed: (**a**) GI % of *Photobacterium damselae* subsp. *piscicida*; (**b**) GI % of *Tericibaculum maritimum*; (**c**) GI % of *Aeromonas. salmonicida* subsp. *salmonicida*; (**d**) GI % of *Edwardsiella tarda* (typical); (**e**) GI % of *Edwardsiella tarda* (atypical) for D60 (I) and NP30 (II). All data are expressed as means ± SEM (*n* = 3).

**Figure 6 microorganisms-11-02877-f006:**
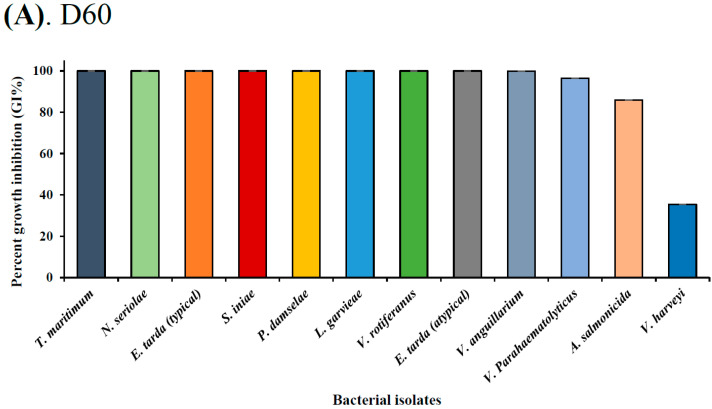
Overall bacterial sensitivity via percent growth inhibition by (**A**) D60 and (**B**) NP30 at 0.5 mg/mL in descending order.

**Figure 7 microorganisms-11-02877-f007:**
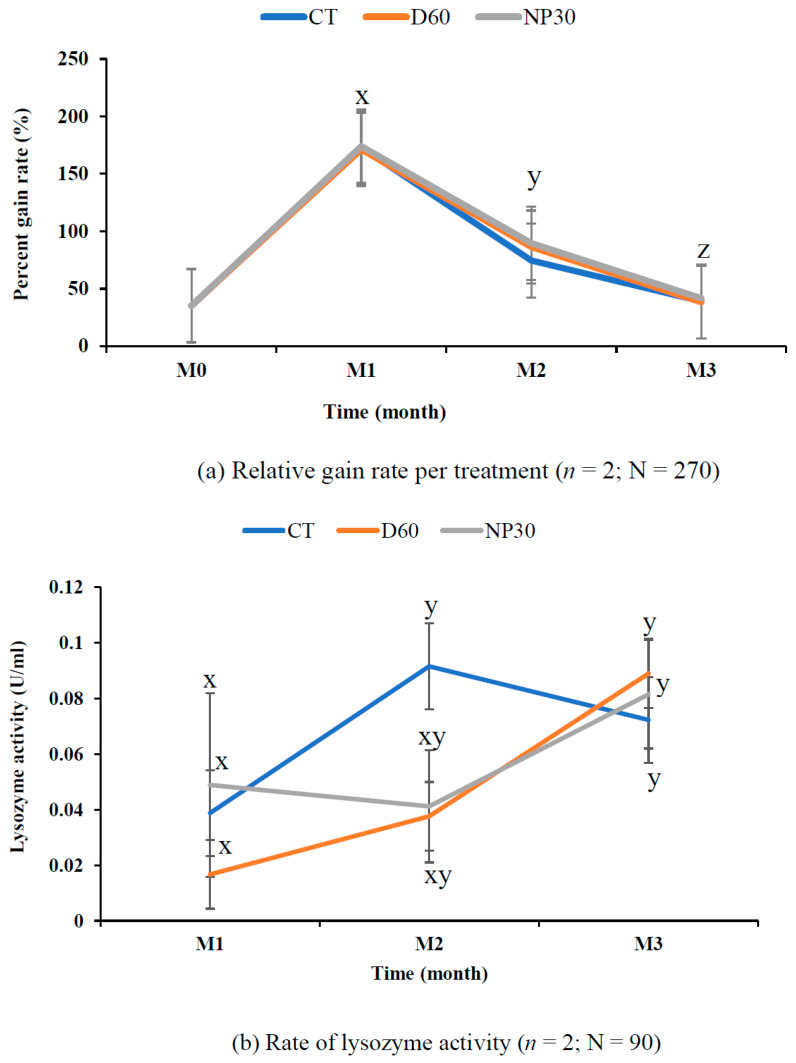
Fish growth and safety parameters. (**a**) Relative gain rate of rainbow trout for the study period. (**b**) Blood plasma lysozyme activity at OD 493 nm. (**c**) Plasma complement activity at OD 493 nm. (**d**) Plasma glucose concentrations. (**e**) Rate of plasma SOD inhibition (%) per treatment. All data are expressed as means ± SEM, where *n* = the number of replicates per treatment and N = the total number of fish for each test. Different letter superscripts, x, y, z, and asterisks (*) indicate significant differences between treatments and or sampling time (*p <* 0.05). CT = control group; M1, M2, and M3 = months 1, 2, and 3.

**Figure 8 microorganisms-11-02877-f008:**
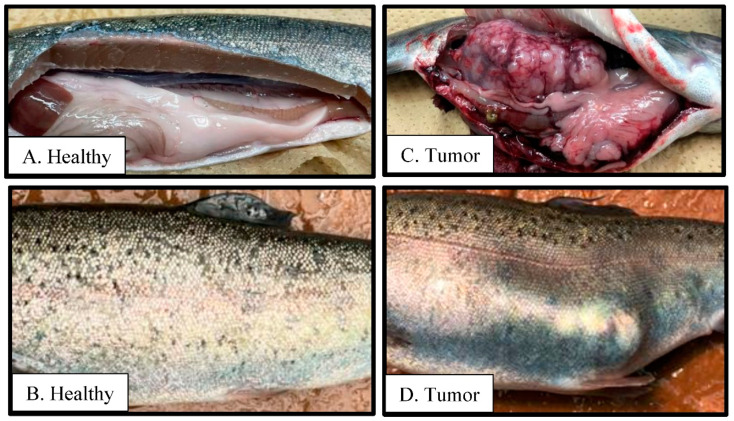
Plate comparing healthy rainbow trout (**A**,**B**) and one with a tumor (**C**,**D**) isolated from an NP30 (N = 1; where N = total number of fish).

**Table 1 microorganisms-11-02877-t001:** Antibacterial activity of isobutyl cyanoacrylate nanoparticles, D60 (150–180 nm) and NP30 (10–50 nm) expressed as MIC and MBC against some fish bacterial pathogens.

Bacteria Type	Species	NP30 (mg/mL)		D60 (mg/mL)	
		MIC	MBC	MIC	MBC
Gram-positive bacteria	*Lactococcus garvieae*	0.156	5.00	0.078	-
	*Streptococcus iniae*	0.078	0.078	0.078	-
	*Nocardia seriolae*	0.313	1.25	0.313	-
Gram-negative bacteria	*Edwardsiella tarda* (typical)	1.25	2.50	5.00	-
	*Edwardsiella tarda* (atypical)	1.25	2.50	5.00	-
	*Aeromonas salmonicida* subsp. *salmonicida*	1.25	2.50	5.00	-
	*Tenacibaculum maritimum*	0.078	0.313	0.078	0.625
	*Photobacterium damselae* subsp. *piscicida*	0.078	5.00	0.078	-
	*Vibrio anguillarium*	0.625	1.25	0.313	2.50
	*V. harveyi*	2.50	2.50	-	-
	*V. parahaemolyticus*	2.50	2.50	5.00	-
	*V. rotiferanus*	2.50	2.50	5.00	-

MIC: minimum inhibitory concentration, MBC: minimum bactericidal concentration, and “-” means the MIC and or MBC value(s) was not obtained at all doses examined in the current study.

**Table 2 microorganisms-11-02877-t002:** Growth performance, feed utilization, and health conditions of rainbow trout fed the test diets (with iBCA-NP—D60, NP30, or without any (control)).

Parameters	Treatment		
	Control	D60	NP30
Initial weight (g)	35.05 ± 0.42	35.07 ± 0.42	35.12 ± 0.52
Final weight (g)	231.75 ± 6.61	244.76 ± 6.46	258.04 ± 7.65
Weight gain (WG %)	561.20 ± 6.62	598.16 ± 40.48	634.78 ± 52.8
Specific growth (% per day)	2.25 ± 0.71	2.31 ± 0.7	2.37 ± 0.69
Feed conversion ratio (FCR)	1.51 ± 0.06	1.25 ± 0.31	1.26 ± 0.26
Survival rate (%)	98.89 ± 1.11	98.89 ± 1.11	100 ± 0.00
Condition factor	1.75 ± 0.02	1.92 ± 0.10	1.83 ± 0.06
Hepatosomatic index	1.03 ± 0.04	1.07 ± 0.05	1.06 ± 0.07
Viscerosomatic index	8.47 ± 0.09	8.46 ± 0.12	8.92 ± 0.46

Results are presented as means ± SEM (*n* = 2). There was no significant difference among treatments for all parameters assessed. Condition factor = 100 × [fish weight/(fish length)^3^]; hepatosomatic index = 100 × (liver weight/fish weight); viscerosomatic index = 100 × (viscera weight/fish weight).

## Data Availability

The data that support the findings of this study are available from the corresponding author upon reasonable request.

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
