# Peer review of "In Vitro Efficacy of Isobutyl Cyanoacrylate Nanoparticles against Fish Bacterial Pathogens and Selection Preference by Rainbow Trout (Oncorhynchus mykiss)"

_microorganisms, 2023, doi:10.3390/microorganisms11122877_

Round 1

Reviewer 1 Report

In this study, D60 and NP30 demonstrated high in vitro bactericidal and or bacteriostatic properties against both the Gram-positive and Gram-negative bacterial pathogens of fish diseases we tested. Although it has shown a good bactericidal effect, the effect in vivo is not clear, and it needs to be further verified that it has been applied to production practice. In fact, it would have been better for the authors to challenge the fish and see how infected they were. On the whole, the content is very meaningful, and it would be better if the prevention and control experiment can be verified.

1. Line 176, Figure 2,From the plot, MIC of NP30 against E. tarda (fl) is 125 μg/ml. in fact, MIC of NP30 against E. tarda (fl) is 1250 μg/ml. please correct it.

2. Figure 5d, II, The maximum value of the vertical should be 100

3. Line 246-251, The format of the formula is confusing and needs to be re-edited

4. The difference letters of Figur 7a,b and c are not easy to distinguish, it is best to re-do the picture or explain clearly.

5. Line 233, mean temperature was 18.56 ± 0.01 oC, How can the temperature be controlled so accurately.

6. Many writing details have not been carefully checked, such as redundant spaces, not fighting for units of writing, etc.

English writing needs to be strengthened. Please find native English speakers to help you revise the language.

Reviewer 2 Report

The manuscript communicates an interesting subject and in general, the study design is sound. My major concern is related to the ethical part of the study. This must be approved by a local Ethics Committee and this is not stated in the text. There is no statistical support or rationale to justify the groups of five animals per experimental condition, this should be clarified by calculating the statistical power. As a minor point, the artwork of figures should be improved and the bar chats format homogenized to include all standard error/deviation bars.

English usage is fine but this can be polished during the manuscript revision.

Round 2

Reviewer 2 Report

I thank the authors for the revised version of the manuscript. I think it is necessary to include the statement that the local ethics committee approved the project, including the reference code. So, my suggestion is to request the document from your local committee.

The analysis of the statistical power can be included in the methodology, with no problem.

There are some typos that can be fixed in the proofreading.
